# Is Continuous Positive Airway Pressure a Valid Alternative to Sildenafil in Treating Sexual Dysfunction among OSA Patients? A Systematic Review and Meta-Analysis

**DOI:** 10.3390/medicina59071318

**Published:** 2023-07-17

**Authors:** Giovanna Stilo, Claudio Vicini, Isabella Pollicina, Antonino Maniaci, Jérôme René Lechien, Christian Calvo-Henríquez, Miguel Mayo Yáñez, Giannicola Iannella, Annalisa Pace, Giovanni Cammaroto, Giuseppe Meccariello, Angelo Cannavicci, Antonio Moffa, Manuele Casale, Ignazio La Mantia

**Affiliations:** 1Department of Medical and Surgical Sciences and Advance and Echnologies “GF Ingrassia”, ENT Section, University of Catania, 95123 Catania, Italy; 2Oral Surgery Unit, Department of Head-Neck Surgery, Otolaryngology, Head-Neck, Morgagni Pierantoni Hospital, 47121 Forli, Italy; claudio@claudiovicini.com (C.V.); a.cannavicci.md@gmail.com (A.C.); 3Research Committee Young-Otolaryngologists of the International Federations of Oto-Rhino-Laryngological Societies (YO-IFOS), 75000 Paris, France; jerome.lechien@umons.ac.be (J.R.L.); christian.ezequiel.calvo.henriquez@sergas.es (C.C.-H.); miguelmmy@gmail.com (M.M.Y.);; 4Department of Human Anatomy and Experimental Oncology, Faculty of Medicine, UMONS Research Institute for Health Sciences and Technology, University of Mons (UMons), 7000 Mons, Belgium; 5Service of Otolaryngology, Hospital Complex of Santiago de Compostela, 15701 Santiago de Compostela, Spain; 6Otorhinolaryngology-Head and Neck Surgery Department, Complexo Hospitalario Universitario A Coruña (CHUAC), 15006 A Coruña, Spain; 7Department of “Organi di Senso”, University “Sapienza”, 00185 Rome, Italy; annalisapace90@gmail.com; 8Unit of Otolaryngology, University Campus Bio-Medico, 00185 Rome, Italy; moffa.antonio1@gmail.com (A.M.); m.casale@unicampus.it (M.C.)

**Keywords:** CPAP, erectile dysfunction, obstructive sleep apnea syndrome

## Abstract

*Background and Objectives*: This study aimed to assess the comparative effectiveness of continuous positive airway pressure (CPAP) therapy and sildenafil pharmacological therapy in improving sexual function among patients with obstructive sleep apnea (OSA) and erectile dysfunction (ED). *Materials and methods*: Population: Patients affected by OSA and ED; Intervention: CPAP therapy vs. Comparison: Sildenafil pharmacological therapy; Outcomes: Improvement in erectile function, as measured by the International Index of Erectile Function 5 (IIEF-5) scoring system; Time: A systematic review of the literature from the past 20 years; Study Design: Observational studies comparing erectile function improvements after OSA treatment. *Results*: A total of eight papers were included in the qualitative summary, involving four hundred fifty-seven patients with ED and OSA. Erectile function improvements were observed in both treatment groups. After sildenafil and CPAP treatment, the mean IIEF-5 domain scores were 37.7 and 27.3, respectively (*p* < 0.001). Sildenafil 100 mg demonstrated a higher therapeutic impact compared to CPAP treatment. *Conclusions*: CPAP therapy significantly improved sexual parameters in most studies for OSA patients with ED. The findings suggest that CPAP therapy effectively alleviates erectile dysfunction symptoms, resulting in improved sexual performance in OSA patients. The comparison of the two treatments indicates that sildenafil has a more substantial therapeutic impact on erectile function than CPAP therapy; however, a combined treatment will provide a cumulative effect.

## 1. Introduction

Obstructive sleep apnea syndrome (OSAS) is a common disorder characterized by repetitive, partial, or complete pharyngeal collapse during the night, leading to sleep fragmentation, intermittent hypoxia, and increased systemic oxidative stress [1]. Furthermore, chronic activation of the sympathetic nervous system contributes to increased cardiovascular and cerebrovascular risk, as well as vascular changes in the peripheral vessel wall [2,3,4]. Several physiopathogenetic mechanisms have been identified in obstructive sleep apnea, leading to small vessel disorders in conditions such as diabetes and neurological and sexual dysfunctions [5].

In 2014, Trzepizur et al. investigated the relationship between microparticles (MPs) and vascular dysfunction in patients with obstructive sleep apnea (OSA) [6]. The authors observed a relationship between the severity of OSA and the levels of circulating MPs, indicating that endothelial and platelet-derived MPs were associated with OSA severity and nocturnal hypoxia, a common feature of OSA. 

Instead, in 2000, Feldman et al. found a significant association between erectile dysfunction and various coronary risk factors, emphasizing the importance of addressing cardiovascular health in the management of erectile dysfunction [7]. Continuous positive airway pressure (CPAP) therapy was demonstrated as effective in improving sleep quality, reducing daytime sleepiness, and potentially mitigating cardiovascular risks associated with obstructive sleep apnea.

In particular, the current literature reports a 48% incidence of sexual problems among men aged 25–65 years with sleep apnea [8,9].

Lisan et al. recently demonstrated accelerated vascular aging in OSA patients, affecting carotid intima-media thickness (β = 0.21; 0.17–0.26), carotid pulse wave velocity (β = 0.31; 0.26–0.35), and carotid diameter (β = 0.43; 0.38–0.48) [10]. Endothelial damage in OSA also involves small caliber vessels through atherosclerosis processes mediated by microparticles (MPs) released by various blood cells containing cytosolic elements [11]. Although the mechanisms underlying the onset of ED in OSAS patients remain unclear, several factors, including impairment of sacral segment function and a lack of nitric oxide (NO), contribute to its development [12,13]. Fanfulla et al. reported early signs of nerve involvement causing ED in patients with OSAS [8]. Conversely, Margel et al. suggested that morning tiredness and the respiratory disturbance index were predictive factors for ED, but only in patients with severe OSAS [9]. Perimenis et al. enrolled 70 patients with ED and demonstrated that both sildenafil 100 mg and CPAP in two separate arms had a positive therapeutic impact [14]. Additionally, Stephan Budweiser et al. showed that long-term CPAP treatment of OSA reduced related intermittent hypoxia, consequently improving moderate to severe erectile dysfunction [15]. The authors demonstrated a certain reversibility of OSA-induced sexual dysfunctions, with the EF subdomain improved in CPAP users vs. non-users (*p* = 0.047). Thus, the relationship between sleep apnea outcomes and patients’ sexual function remains variable and controversial in the literature, with a lack of scientific evidence analyzing the actual impact, as in other areas, on sleep-related respiratory disorders [16,17]. Therefore, we carried out a systematic review and meta-analysis to determine the effectiveness of sildenafil versus CPAP for patients with ED and OSA.

## 2. Materials and Methods

### 2.1. Systematic Review Protocol, Data Extraction, and Evaluation of Outcomes

We adhered to the Cochrane guidelines to ensure a rigorous and transparent process [18]. Firstly, we defined a clear research question focusing on the effects of CPAP treatment and sildenafil on erectile dysfunction in patients with obstructive sleep apnea. We then developed a protocol outlining our objectives, methods, and planned analyses, which was registered in the PROSPERO database. To identify relevant studies, we conducted a comprehensive search of multiple electronic databases, such as PubMed/Medline, Embase, Web of Science, Scholar, and the Cochrane Library, using a combination of keywords and search filters. 

By following the Cochrane guidelines throughout our systematic review, we ensured the production of high-quality, reliable evidence to inform clinical decision-making regarding the treatment of erectile dysfunction in patients with obstructive sleep apnea.

Two authors (A.M and G.S) screened the titles, abstracts, and full-text articles to select eligible studies based on pre-specified inclusion and exclusion criteria. Data extraction was performed using a standardized form, with any discrepancies resolved through discussion or a third reviewer. Any disagreements were resolved through discussion among study members. All included studies were reviewed to obtain all available data, ensuring the eligibility of all subjects. So, the main characteristics of patients, symptoms, diagnostic procedures, treatment modalities, outcome scores (AHI, apnea–hypopnea index; ESS, Epworth sleepiness scale; ODI, oxygen desaturation index; LOS, lower oxygen saturation) and follow-up. CPAP treatment and sildenafil erectile dysfunction outcomes in OSA patients were retrieved and evaluated.

### 2.2. Electronic Database Search

In accordance with the PRISMA checklist for reviews and meta-analyses, we performed a systematic review of the current literature through the major online databases PubMed/Medline, Embase, Web of Science, Scholar, and the Cochrane Library (Figure 1).

The primary outcome searched was the change in erectile dysfunction after medical treatment with CPAP or sildenafil in patients with OSA over the past 20 years (1 January 2002 to 1 July 2022) by two different authors. The following search keywords were used: “CPAP treatment” and “erectile dysfunction”; “continuous positive airway pressure” and “erectile dysfunction”; “sildenafil” and “erectile dysfunction”; “obstructive sleep apnea” and “erectile dysfunction”; “obstructive sleep apnea”, “IIEF-5”, and “CPAP”; and “IIEF-5” and “sildenafil”. We also considered the option “related articles option” on the home pages of PubMed and Scholar. All investigators reviewed the titles and abstracts of articles available in English. Then, the identified full-text articles were screened for original data and were manually checked and retrieved for other relevant studies.

### 2.3. Inclusion and Exclusion Criteria

The included studies met the following criteria:Original article comparing pre-treatment and post-treatment (CPAP or sildenafil) outcomes of erectile dysfunction in patients with OSA.The article was published in English.The studies performed medical treatment after confirming obstructive sleep apnea on polysomnography (PSG).All studies reported detailed information on pre-operative and post-treatment respiratory indices (PSG), such as AHI, ESS, oxygen desaturation, ODI, and LOS.

We excluded:Case reports, editorials, letters to the editor, or reviews.Studies with only qualitative outcomes.Patients with central or mixed sleep apnea.Papers lacking continuous pre- and post-treatment data.

Furthermore, all studies were considered ineligible if they had not carried out an adequate selection of patients, such as, for example, not reporting a sample age variable, not reporting a diagnostic confirmation of OSA established by polysomnography (PSG) or home sleep apnea test (HSAT) or having an apnea–hypopnea index (AHI) greater than five events per hour. Moreover, studies were considered inadequate if they did not report assessed erectile dysfunction using the International Index of Erectile Function (IIEF) validated questionnaire.

### 2.4. Statistical Analysis

We performed the search protocol according to the validated reporting items’ quality requirements for systematic review and meta-analysis protocols (PRISMA) declaration [19]. Furthermore, the study quality assessment (Joanna Briggs Institute) tool was used to evaluate the study design features of the included articles [20]. Statistical analysis was performed using statistical SPSS software (IBM SPSS Statistics for Windows, IBM Corp. Released 2017, Version 25.0. Armonk, NY, USA: IBM Corp). Furthermore, we adopted random-effects modeling (standard error estimate = inverse of the sample size) to estimate the summary effect measures by 95% confidence intervals (CI); subsequently, forest plots were generated via the Review Manager Software (REVMAN) version 5.4 (Copenhagen: The Nordic Cochrane Centre: The Cochrane Collaboration). Thus, the inconsistency (I^2^ statistic) was calculated, and the values for low inconsistency = 25%, moderate inconsistency = 50%, and high inconsistency = 75% were established. The calculation of the optimal total sample size was conducted using the G-Power statistical software. An alpha error of 0.05, an average effect size of 0.50, and a power greater than 85% were foreseen based on date in the literature. To reduce clinical heterogeneity, studies with an overall sample size of fewer than ten patients were excluded from the analysis. The protocol review was registered in the PROSPERO database (cod. 364190). 

## 3. Results

### 3.1. Retrieving Researches

We identified 315 potentially relevant studies through the systematic review of the literature (Figure 1). After removing the duplicates and applying the criteria listed above, an overall number of 191 records screened were potentially relevant to the topic. We excluded all the studies that did not match the inclusion criteria through the records analysis and the following articles’ full-text screening. Thus, the remaining eight papers were included in qualitative synthesis papers for data extraction. After the meta-analysis established the criteria, we considered eight studies for quantitative analysis. The potential risk of bias in observational studies was assessed using the Joanna Briggs Institute Critical Assessment Checklist for Observational Studies (Figure 2).

According to the GRADE assessments, the risk of bias was mostly unclear, inconsistency was predominantly moderate, indirectness was largely direct, imprecision was generally present, and publication bias was largely unclear (Appendix A). The overall quality of the evidence was mixed, with several low-quality studies and a few moderate-quality studies. 

### 3.2. Patients’ Features and Medical Treatment

We have included eight articles [19,20,21,22,23,24,25,26] in our systematic literature review. Out of a total of 457 OSAS patients with erectile dysfunction, 381 (83.36%) were treated with CPAP [19,20,21,22,23,24,25,26] and 76 (16.63%) with sildenafil [21,22,25] (Table 1).

The average age of patients was 48.3 ± 7.4 y for patients treated with CPAP and 48.2 ± 11.4 y for patients treated with sildenafil. All participants in the studies were male subjects. All participants in the studies had undergone a home sleep apnea test, polysomnography type III (HSAT), as defined in the AASM rules [21]. All subjects were evaluated according to age, body mass index (BMI), AHI, IIEF-5, and ESS scores. 

The pooled BMI of the patients was 29.94 ± 3.02, of which, at subgroup analysis, 29.94 ± 3.02 for CPAP and 25.5 ± 4.1 for sildenafil. 

Two papers (120 patients) [23,24] analyzed the ESS outcomes after CPAP treatment, demonstrating a significant reduction from 11.88 ± 1.82 to 6.2 ± 1.48 (*p* < 0.001).Three papers analyzed the AHI outcomes after CPAP treatment, demonstrating a significant reduction from 32.24 ± 24.96 to 10.67 ± 2.51 (*p* < 0.01); LOS were reported in three papers, with a significant score improvement from the value of 78.8 ± 7.9 to 89.4 ± 3.9 (*p* < 0.01) [23,25,26]. At pooled analysis for EF outcomes, both medical treatments demonstrated a higher EF score, but sildenafil reported greater outcomes than CPAP (mean: CPAP 27.3 vs. sildenafil 37.7; (*p* < 0.01) [20,21,24].

### 3.3. CPAP Treatment

The results were obtained from a total of eight papers, which included three hundred eighty-one patients who received continuous positive airway pressure (CPAP) treatment. These studies assessed erectile function using the International Index of Erectile Function (IIEF) and the Erectile Dysfunction Inventory of Treatment Satisfaction (EDITS) questionnaires. The patients’ apnea–hypopnea indexes (AHIs) were measured both before (AHI > 30 events/h) and after treatment (AHI > 20 events/h). In patients who underwent CPAP therapy, erectile function (EF) scores following treatment were significantly higher compared to the pre-treatment scores (mean: 27.3 vs. 7; *p* < 0.001). A random-effects model was employed to analyze the results for the three hundred eighty-one patients treated with CPAP across the eight papers. The analysis demonstrated a mean difference (MD) of −2.98 in AHI EF scores (95% confidence interval (CI) −6.38, 0.41). The overall effect Z score for CPAP was 1.72 (*p* = 0.09), indicating a trend towards improved erectile function following treatment. However, the Q statistic *p*-value was < 0.001, suggesting statistically significant heterogeneity among the studies, and the I² value was 97%, indicating a high level of inconsistency in the results, as depicted in Figure 3.

These findings imply that CPAP treatment may lead to improvements in erectile function for patients with obstructive sleep apnea. However, the high level of heterogeneity among the studies should be considered when interpreting the outcomes, standardized methodology, and rigorous study designs.

### 3.4. Sildenafil vs. CPAP Treatment

The outcomes after medical treatment were reported in three papers, which included a total of one hundred fifty-one patients. All of these papers utilized the same drug, sildenafil, at a consistent dosage of 100 mg for the treatment of erectile dysfunction (ED) in patients with obstructive sleep apnea (OSA). The outcomes demonstrated that in patients treated with sildenafil, the erectile function (EF) scores after treatment were significantly higher than before treatment (mean: 37.7 vs. 7.8; *p* < 0.001) (Figure 4).

These findings suggest that sildenafil administration at a 100 mg dosage effectively improved erectile function in OSA patients, with a substantial increase in EF scores highlighting the potential benefits of using sildenafil as a treatment option to address ED associated with OSA.

After treatment, the group treated with sildenafil showed a significantly higher mean score of erectile function (EF) than the group treated with continuous positive airway pressure (CPAP) (mean: 37.7 vs. 27.3; *p* < 0.001). This suggested that sildenafil may be more effective in improving erectile function in patients with obstructive sleep apnea (OSA) compared to CPAP alone. Furthermore, the analysis using random-effects modeling for seventy-six patients treated with sildenafil (involving three papers) demonstrated a mean difference of −5.97 in apnea–hypopnea index (AHI) EF scores (95% confidence interval (CI) −8.88, −3.05). This overall effect had a Z score of 4.01, indicating a statistically significant improvement in erectile function with sildenafil treatment. The Q statistic *p*-value was 0.09, suggesting the presence of statistically significant heterogeneity among the studies. The I² value was 58%, which indicates a moderate level of heterogeneity in the analysis. These results, as depicted in Figure 5, supported the notion that sildenafil treatment may lead to a greater improvement in erectile function for patients with OSA compared to CPAP treatment alone. However, it is essential to consider the limitations of the study and the potential influence of heterogeneity among the included studies when interpreting these findings.

## 4. Discussion

To date, our study is the first comprehensive systematic review and meta-analysis focused on erectile function disorders in 381 OSA patients, compiling and evaluating the current evidence on the potential benefits of CPAP treatment. Our findings indicate that CPAP therapy may result in improved erectile function; however, the significant heterogeneity among the studies necessitates caution in interpreting the data. Specifically, in patients receiving CPAP treatment, post-treatment erectile function (EF) scores exhibited a trend towards enhanced erectile function, with notably higher scores compared to pre-treatment (mean: 27.3 vs. 7; *p* < 0.001) and a Z-effect score of 1.72 (*p* = 0.09). Furthermore, we also noted that single-treatment CPAP cannot yet supplant sildenafil as the preferred treatment for erectile dysfunction. In a subgroup analysis, the sildenafil-treated subjects demonstrated a significantly higher mean score than the CPAP ones (mean: 37.7 vs. 27.3; *p* < 0.001).

Several studies have reported an association between erectile dysfunction (ED) and obstructive sleep apnea (OSA) [21,26,27,28]. 

Different etiopathogenic theories are imputed to ED in OSA, including repeated episodes of intermittent hypoxia and hyperadrenergic responses, reduced NO levels, and chronic endothelial damage.

The pathophysiological mechanisms underlying ED in OSA are hypothesized to involve repeated nocturnal hypoxia, increased sympathetic activity, reduced nitric oxide levels, and endothelial damage [1,2,3,4,5,6,10,11,12,13]. Specifically, greater severity and duration of desaturation episodes during sleep are correlated with worse ED, especially at oxygen saturations < 80% [26]. Reduced saturation is correlated with DE, with greater dysfunction at SaO2 < 80% [26]. Different studies have investigated the efficacy of CPAP treatment in ED, reporting mixed results [19,20,21,22,23,24,25,26]. CPAP could improve ED through increased circulating NO levels and improved endothelial function [20]. Pascual et al. [26] found that 3 months of CPAP improved ED and sexual satisfaction in 35 patients with moderate to severe OSA. They reported increased nocturnal penile tumescence and improved EF based on the International Index of Erectile Function after CPAP. 

After CPAP, patients could experience performance improvement related both to subjective outcomes such as depression and anxiety scores or vascular and inflammatory ameliorations compared to baseline. Sforza et al. [1] found that chronic intermittent hypoxia in mice and OSA patients led to increased reactive oxygen species, inflammation, and apoptosis in the penis, which are mechanisms that could contribute to ED. 

May et al. [2] proposed that hypoxia-induced endothelial dysfunction and increased sympathetic activity in OSA alter cavernosal smooth muscle tone and arterial inflow to the penis. Instead, Ramar et al. [3] described OSA as an independent risk factor for endothelial dysfunction and accelerated atherosclerosis that may impair erectile function.

Several studies have examined the effects of continuous positive airway pressure (CPAP) on ED in OSA patients, with mixed results [21,22,23,24,25,26,27,28]. CPAP may improve erectile function by reducing nocturnal hypoxia, decreasing sympathetic activation, increasing nitric oxide levels, and restoring endothelial function [22]. Our meta-analysis of eight studies found a significant mean improvement of 27.3 in erectile function (EF) scores after CPAP treatment (*p* < 0.001). Pooled analysis using random-effects modeling showed a mean difference of −2.98 (95% CI −6.38, 0.41) in apnea–hypopnea index EF scores before and after CPAP, with significant heterogeneity (*p* < 0.001, I^2^ = 97%). The effect of CPAP was also confirmed in both the different severity classes of mild or severe OSAS and at short- or long-term follow-ups [20,22,24,25].

Perimenis et al. [21] found that 3 months of CPAP significantly improved EF scores in men with severe OSA. Khafagy et al. [23] reported improved ED as early as 1 month after CPAP initiation, with further gains over 6–12 months. Taskin et al. [24] found that CPAP for at least 4 h per night for 3 months significantly improved ED, especially in younger patients with severe OSA and hypoxia. Pascual et al. [26] showed that 3 months of CPAP improved ED and sexual satisfaction while reducing depression scores. Coban et al. [27] reported improved EF, lower urinary tract symptoms, and better quality of life after 3 months of CPAP in severe OSA.

The phosphodiesterase type 5 inhibitor, sildenafil, is a mainstay of treatment for ED when an underlying cause is not found [19]. In this context, CPAP usage could produce a cumulative positive effect on sildenafil treatment for EF in OSA patients. Indeed, Perimenis et al. [22] found that sildenafil plus CPAP was superior to sildenafil alone for ED in OSA patients.

The long-term outcomes were instead assessed by Mittleman et al. [17], using on-demand sildenafil citrate over 1–2 years in 6293 men. In a subgroup of patients with ED secondary to comorbid conditions such as diabetes or OSA (*n* = 925, 15%), sildenafil significantly improved erection-based events with promising prospectives.

In accordance with the evidence presented, our pooled analysis of three studies (*n* = 76) using sildenafil 100 mg found a significant mean improvement of 37.7 in EF scores (*p* < 0.001) and a mean difference of −5.97 (95% CI −8.88, −3.05, *p* = 0.09, I^2^ = 58%) favoring sildenafil.

Moreover, in the analysis using random-effects modeling for seventy-six patients, sildenafil-treated patients (three papers) demonstrated a mean difference of −5.97 [95% CI −8.88, −3.05], favoring sildenafil treatment, an overall effect Z score = 4.01, Q statistic *p* = 0.09 (statistically significant heterogeneity), and I^2^ = 58%.

Despite these encouraging findings, our study is limited by potential selection bias from variable control of comorbidities such as hypertension or diabetes mellitus that can also cause ED. The small number of studies and sample sizes also necessitate further randomized controlled trials to compare CPAP and sildenafil for treating ED in OSA patients, as reported by the grade assessment.

Moreover, it is essential to emphasize the role of common cardiovascular comorbidities in patients with OSAS, including obesity, hypertension, and frequent contributing factors such as diabetes and smoking. These factors can indeed influence the vasculogenic nature of ED, representing crucial variables to consider when examining the relationship between OSAS and ED.

The lack of proper analysis of the influence of these factors and the absence of stratification based on each variable in the meta-analysis may have led to an incomplete understanding of the relationship between OSAS and ED. By not accounting for the potential impact of comorbidities such as obesity, hypertension, diabetes, and smoking, the study results may not accurately represent the true effects of CPAP treatment of erectile function in the context of these contributing factors.

A more thorough investigation into the role of these comorbidities and appropriate stratification could provide a clearer picture of how CPAP treatment may affect erectile function in patients with OSAS. This additional analysis would also help to identify potential confounders and improve the overall quality of the study. To obtain a more accurate understanding of the relationship between OSAS and ED, future research should address these limitations by incorporating stratification based on various comorbidities and conducting in-depth analyses of the effect of each contributing factor on the study outcomes.

One such limitation is the possibility of publication bias, as studies with positive or significant findings are more likely to be published than those with negative or insignificant results. This could lead to an overestimation of the treatment effects of CPAP and sildenafil.

Similarly, OSAS has been associated with other relevant conditions such as nocturia and lower urinary tract symptoms (LUTS), which should also be incorporated into future research as they could help increase the comprehensiveness of the evidence [28,29].

A study by Clerget et al. investigated the effects of continuous positive airway pressure (CPAP) therapy on LUTS in male patients with OSAS [28]. The results demonstrated that CPAP treatment significantly improved LUTS, including nocturia, in these patients, suggesting a potential link between the two conditions.

In the same vein, a study by Irer et al. examined the prevalence of LUTS and the association with OSAS severity in a large sample of male patients [14]. They found that patients with severe OSAS had a higher prevalence of LUTS, including nocturia, further supporting the association between OSAS and these bothersome symptoms.

A recent study from Di Bello et al. explored the relationship between OSAS and LUTS, including nocturia, and found a significant correlation between the severity of OSAS and the presence of these symptoms [29]. The authors suggested that proper management of OSAS might lead to improvements in LUTS and nocturia.

Another limitation is the significant heterogeneity among the included studies in terms of study design, patient populations, and treatment durations. This could affect the overall conclusions drawn from the meta-analysis. Although random-effects modeling was used to account for some of this heterogeneity, it still poses a challenge to the study.

Furthermore, many of the included studies did not implement blinding or placebo control, which could introduce bias and affect the internal validity of the findings. The use of various methods to assess erectile function, such as self-report questionnaires and objective measures like nocturnal penile tumescence, also complicates the comparison of results across studies due to inconsistencies in the reported outcomes.

Lastly, the follow-up duration in many of the included studies was relatively short, limiting the evaluation of the long-term treatment effects of CPAP and sildenafil on erectile function in OSA patients.

## 5. Conclusions

This meta-analysis concentrates on recent findings regarding the treatment of erectile dysfunction (ED) in patients with obstructive sleep apnea. While continuous positive airway pressure (CPAP) therapy has demonstrated promising outcomes in reducing ED, to date, sildenafil as a standalone treatment has yielded fewer encouraging results. Furthermore, although a combination therapy of sildenafil and CPAP could be hypothetically suggested, there are currently no prospective studies examining their combined effects. Consequently, additional trials are necessary to determine the validity and efficacy of one treatment compared to the other.

## Figures and Tables

**Figure 1 medicina-59-01318-f001:**
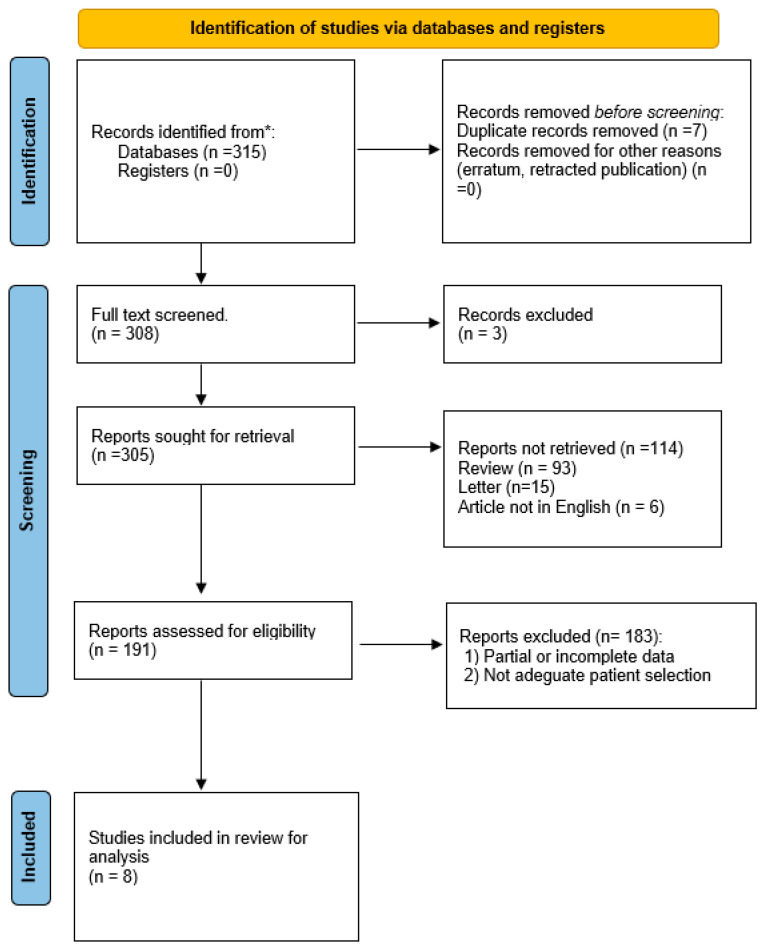
PRISMA flow diagram. * Paper initially identified.

**Figure 2 medicina-59-01318-f002:**
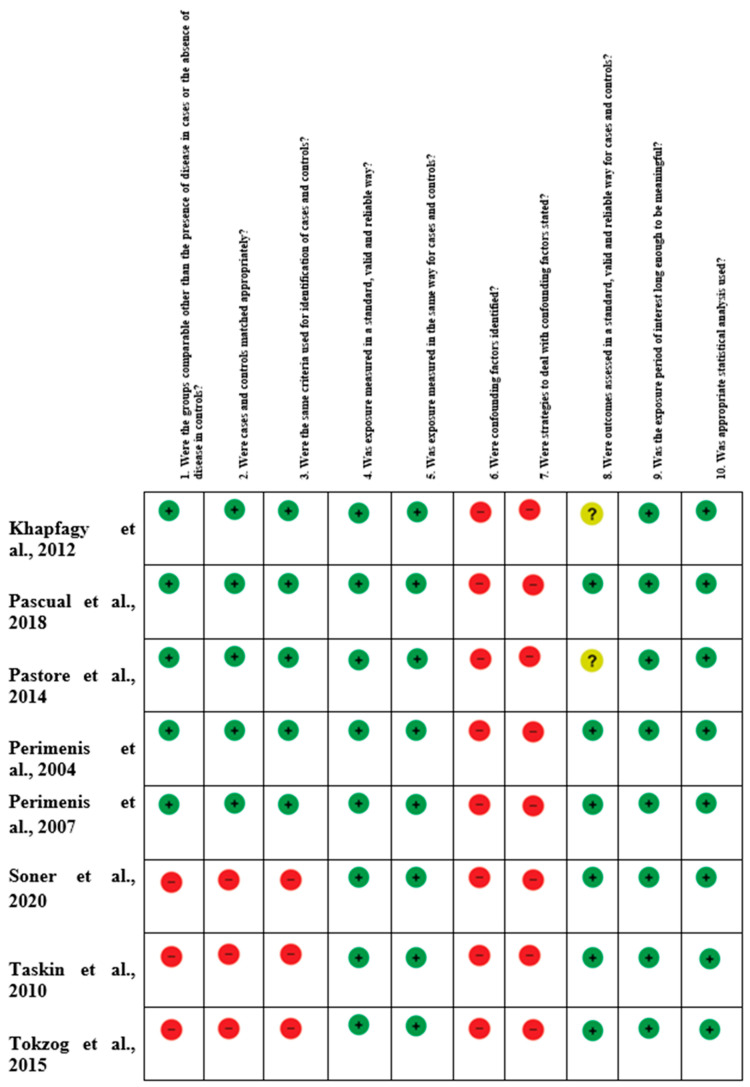
Risk of bias summary [20,21,22,23,24,25,26,27]. The author’s judgments for each included study, assessed by the Joanna Briggs Institute (JBI) Critical Assessment Checklist for Observational Studies.

**Figure 3 medicina-59-01318-f003:**
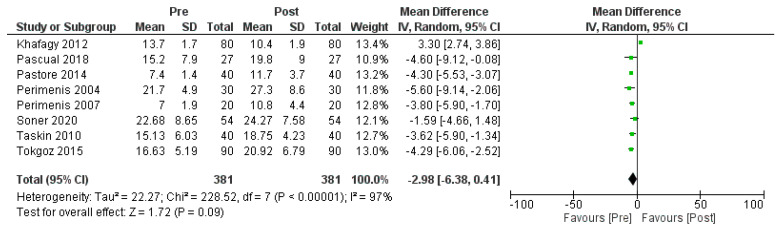
CPAP treatment outcomes on erectile dysfunction [20,21,22,23,24,25,26,27]. Green box: point of estimate. Black Diamond: average point of estimate of all the studies included.

**Figure 4 medicina-59-01318-f004:**
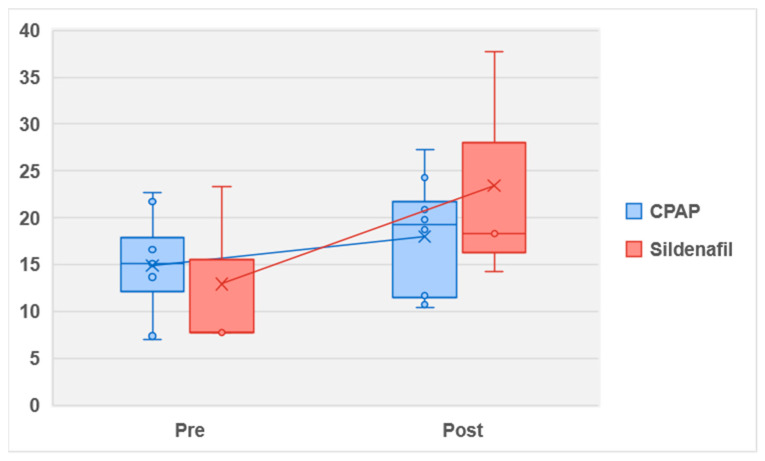
Boxplot pre- vs. post-treatment for EDS outcomes.

**Figure 5 medicina-59-01318-f005:**
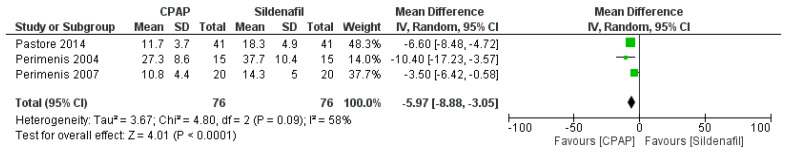
Forest plot of CPAP vs. sildenafil on sexual dysfunction. Green box: point of estimate. Black Diamond: average point of estimate of all the studies included [20,21,24].

**Table 1 medicina-59-01318-t001:** Main demographic features of the study included. Abbreviations: Control, no treatment; AHI, apnea–hypopnea index.

References	Study Design	Sample	Pre-CPAP (Mean, SD)		Post-CPAP (Mean, SD)		Pre-Sildenafil (Mean, SD)		Post-Sildenafil (Mean, SD)		AHI Severity
Perimenis et al., 2007 [20]	Prospective uncontrolled	40 (CPAP 20 vs. Sildenafil 20)	7	1.9	10.8	4.4	7.8	1.2	14.3	5	>30e/h
Perimenis et al., 2004 [21]	Prospective uncontrolled	30 (CPAP 15 vs. Sildenafil 15)	21.7	4.9	27.3	8.6	23.3	2.7	37.7	10.4	>30 e/h
Khafagy et al., 2012 [22]	Prospective uncontrolled	80 (40 CPAP 40 control)	13.7	1.7	10.4	1.9					between 5 and 15 e/h
Taskin et al., 2010 [23]	Prospective controlled	40 (CPAP 20 vs. control 20)	15.13	6.03	18.75	4.23					>30 e/h
Pastore et al., 2014 [24]	Prospective controlled	82 (CPAP 41 vs. Sildenafil 41)	7.4	1.4	11.7	3.7	7.8	1.2	18.3	4.9	>30 e/h
Pascual et al., 2018 [25]	Prospective controlled	60 (CPAP 27 vs. control 30)	15.2	7.9	19.8	9					>20 e/h
Soner et al., 2020 [26]	Prospective uncontrolled	54 CPAP	22.68	8.65	24.27	7.58					>30 e/h
Tokgoz et al., 2015 [27]	Prospective uncontrolled	90 CPAP	16.63	5.91	20.92	6.79					>20 e/h

## Data Availability

Not applicable.

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
