# Peer review of "Is Continuous Positive Airway Pressure a Valid Alternative to Sildenafil in Treating Sexual Dysfunction among OSA Patients? A Systematic Review and Meta-Analysis"

_medicina, 2023, doi:10.3390/medicina59071318_

Round 1

Reviewer 1 Report (Previous Reviewer 1)

The authors made all the comments in the right way.

Author Response

Dear editor your help was precious to improve the paper.

Best regards.

Reviewer 2 Report (Previous Reviewer 2)

Justify how conclusion states that combination will be more effective when this was not studied in the authors work. Consider removing if you cannot justify.

Paragraph 1 of introduction albeit better is still numerous studies relevant to ED and OSA listed in a disorganized manner. Kindly revise to convey a unifying message.

First paragraph of the discussion should summarize and discuss your results. Consider revising discussion.

Author Response

Dear reviewer,

Thanks for the precious help improving the paper.

Comment: Justify how conclusion states that combination will be more effective when this was not studied in the authors work. Consider removing if you cannot justify.

Response: dear reviewer we modified the conclusions according to your suggestions. In particular we highlighted that missing data are present on combined treatment with cpap and sildenafil and more evidences are needed: below the new conclusions:

This meta-analysis concentrates on recent findings regarding the treatment of erectile dysfunction (ED) in patients with obstructive sleep apnea. While continuous positive airway pressure (CPAP) therapy has demonstrated promising outcomes in reducing ED, to date, sildenafil as a standalone treatment has yielded fewer encouraging results. Furthermore, although a combination therapy of sildenafil and CPAP could be hypothetically suggested, there are currently no prospective studies examining their combined effects. Consequently, additional trials are necessary to determine the validity and efficacy of one treatment compared to the other.

Comment: Paragraph 1 of introduction albeit better is still numerous studies relevant to ED and OSA listed in a disorganized manner. Kindly revise to convey a unifying message.

Response: dear reviewer, thanks for the suggestions. We’ve discussed to better unifiy the two papers reported, separating the citations. It wa salso required by the editor to improve the minimum word count of the paper because not sufficient. We hope that now is more clear and unified for the reader. Below the 2 studies discussed:

 Trzepizur et al. in 2014 investigated the relationship between microparticles (MPs) and vascular dysfunction in patients with obstructive sleep apnea (OSA)[6]. The authors observed a relationship between the severity of OSA and the levels of circulating MPs, indicating that endothelial and platelet-derived MPs were associated with OSA severity and nocturnal hypoxia, a common feature of OSA.

Instead, Feldman et al.'s in 2000 found a significant association between erectile dys-function and various coronary risk factors, emphasizing the importance of addressing cardiovascular health in the management of erectile dysfunction[7]. Continuous posi-tive airway pressure (CPAP) therapy was demonstrated effective in improving sleep quality, reducing daytime sleepiness, and potentially mitigating cardiovascular risks associated with obstructive sleep apnea.

  1. Trzepizur, W.; Martinez, M.C.; Priou, P.; et al. Microparticles and vascular dysfunction in obstructive sleep apnoea (2014). Eur Respir J.44(1):207-216. doi:10.1183/09031936.00197413
  2. Feldman, H.A.; Johannes, C.B.; Derby, C.A.; et al. Erectile dysfunction and coronary risk factors: prospective results from the Massachusetts male aging study(2000). Prev Med 30(4):328-338. doi:10.1006/pmed.2000.0643

Comment: First paragraph of the discussion should summarize and discuss your results. Consider revising discussion.

Response: Dear reviewer thanks for the suggestions, we’ve reformulated the first paragraph of discussion to introduce the mayors findings of the paper. Below the text.

To date, our study is the first comprehensive systematic review and meta-analysis focused on erectile function disorders in 381 OSA patients, compiling and evaluating the current evidence on the potential benefits of CPAP treatment. Our findings indicate that CPAP therapy may result in improved erectile function; however, the significant heterogeneity among the studies necessitates caution in interpreting the data. Specifically, in patients receiving CPAP treatment, post-treatment erectile function (EF) scores exhibited a trend towards enhanced erectile function, with notably higher scores compared to pre-treatment (mean: 27.3 vs. 7; p < 0.001) and a Z-effect score of 1.72 (p = 0.09). Furthermore, we also noted that single-treatment CPAP cannot yet supplant sildenafil as the preferred treatment for erectile dysfunction. In a subgroup analysis, the sildenafil-treated subjects demonstrated a significantly higher mean score than the CPAP ones (mean: 37.7 vs. 27.3; p < 0.001).

Best regards.

Reviewer 3 Report (New Reviewer)

Authors should be congratulated for their work. OSA represents a health concern with an underestimated prevalence due to the cross-sectional symptoms and the comorbidities of OSA patients to whom we ascribed most of the symptoms experienced. The manuscript presented a well-structured and solid methodology but Table 1 is so challenging to read. Moreover, several points of discussion must be clarified to improve the overall manuscript quality:

- First, the discussion section did not match the point. Specifically, OSAS patients were generally obese, hypertensive, usually diabetics and/or smokers, and/or with heart disease.  In the manuscript, the vasculogenic nature of ED in OSAS patients, worsened by cardiovascular risk factors co-presence is not stressed enough. Furthermore, it is a not-improvable bias selection that could be better discussed in the limitation section.

- Second, OSAS have related to several bothering conditions such as nocturia or LUTS. This relation was previously described and the manuscript may clarify and include these findings, to increase the comprehensiveness of the evidence cited (PMID: 32830023, 30118776, 37167825).  

- Authors should clarify "not adequate patients selection" presented in the PRISMA figure, in the materials and methods section. 

The language is full of Italianisms that must be correct, such as "Numerous" in the Discussion section. 

Author Response

Authors should be congratulated for their work. OSA represents a health concern with an underestimated prevalence due to the cross-sectional symptoms and the comorbidities of OSA patients to whom we ascribed most of the symptoms experienced. The manuscript presented a well-structured and solid methodology

Dear reviewer thanks for your help and the suggestions required. We’ve improved the paper accordingly.

Comment: Table 1 is so challenging to read.

Response: thanks for the indications, we’ve modified all the table, both in structure,number values and captation to make it more clear.

Comment:  First, the discussion section did not match the point. Specifically, OSAS patients were generally obese, hypertensive, usually diabetics and/or smokers, and/or with heart disease.  In the manuscript, the vasculogenic nature of ED in OSAS patients, worsened by cardiovascular risk factors co-presence is not stressed enough. Furthermore, it is a not-improvable bias selection that could be better discussed in the limitation section.

Response: thanks for the suggestions, we critically analyzed the role of the comorbidities leading to a selection bias for the findings reported.

Comment: Second, OSAS have related to several bothering conditions such as nocturia or LUTS. This relation was previously described and the manuscript may clarify and include these findings, to increase the comprehensiveness of the evidence cited (PMID: 32830023, 30118776, 37167825).  

Response: dear reviewer as below we discussed the role of these evidences as comorbidities that influence the results of osa patiens with EF, especially as limitations of the findings reported in our study. Below the new sentences:

Similarly, OSAS has been associated with other relevant conditions such as nocturia and lower urinary tract symptoms (LUTS), which should also be incorporated into future research as they could help increase the comprehensiveness of the evidence [28-30].

A study by Clerget et al. investigated the effects of continuous positive airway pressure (CPAP) therapy on LUTS in male patients with OSAS [28]. The results demonstrated that CPAP treatment significantly improved LUTS, including nocturia, in these patients, suggesting a potential link between the two conditions.

In a similar vein, a study by Irer et al. examined the prevalence of LUTS and its associ-ation with OSAS severity in a large sample of male patients [29]. They found that pa-tients with severe OSAS had a higher prevalence of LUTS, including nocturia, further supporting the association between OSAS and these bothersome symptoms.

Lastly, a recent study from Di Bello et al. explored the relationship between OSAS and LUTS, including nocturia, and found a significant correlation between the severity of OSAS and the presence of these symptoms [30]. The authors suggested that proper management of OSAS might lead to improvements in LUTS and nocturia.

Comment:  Authors should clarify "not adequate patients selection" presented in the PRISMA figure, in the materials and methods section. 

Response: dear reviewer, we’ve reported other eligibility inclusion referring to the inadequate inclusion into the analysis. Below the concept:

Furthermore, all studies were considered ineligible that have not carried out an adequate selection of patients such as, for example, not reporting a sample age variable, not re-ported a diagnostic confirmation of OSA established by polysomnography (PSG) or home sleep apnea test (HSAT) or an apnea-hypopnea index (AHI) greater than 5 events per hour. Moreover, studies were considered inadequate if did not report an assessed erectile dysfunction using the International Index of Erectile Function (IIEF) validated questionnaire.

Comments: The language is full of Italianisms that must be correct, such as "Numerous" in the Discussion section. 

Response: thanks for the suggestions, we’ve corrected all the italianism.

Bests

Round 2

Reviewer 3 Report (New Reviewer)

Authors answered properly to all suggestions

This manuscript is a resubmission of an earlier submission. The following is a list of the peer review reports and author responses from that submission.

Round 1

Reviewer 1 Report

Dear Editor, and

Dear Authors

This is a very interesting topic about the clinical response on erectile dysfunction in OSA patients with two different therapeutic modalities – sildenafil and CPAP therapy.

The authors did not use the appropriate methodology according to Cochrane and Grade for a systematic reviewing process and meta-analysis. The risk of bias was assessed via the JBI Checklist for case control studies, which is not appropriate for the population studied.

Τhe effect estimates presented in the results section and figures are contradictory. Specifically, the authors investigated the effect estimate of CPAP treatment on the AHI score, which is not an outcome of interest for this review, as CPAP is already approved for OSA treatment. On the contrary, figure 3 states the impact on erectile dysfunction using the same effect estimate. Furthermore the forest plot shows favor pre-treatment.

Also, per protocol, title and methodology, the authors should demonstrate a comparison of CPAP and sildenafil instead of pre to post CPAP treatment.  

Again, in figure 5 they describe the effect of sildenafil and CPAP on sexual dysfunction, while in the corresponding text the effect estimate refers to the AHI score, where sildenafil is not an appropriate treatment.

The quality of the presented evidence was not rated according to Grade.

There were also numerous minor aspects, as noted bellow.  

Minor Comments

Line 46. Please correct “Lisa et al.” to Lisan et al.

Line 71. Please provide in this lines the explanation of the abbreviations AHI, ESS,ODI, LOS and not to the Lines 97-99.

Lines 161-162. Please re-write this sentence as there no meaning.

Lines 222-223. This sentence uses the same words (combine) and must be re-wrote as included in the Conclusion and gives the final judgment.

Table 1. Please replace from Italian to English the phrase “Cpap pre gruppo atttivo”.

Reviewer 2 Report

C1. Abstract should be rewritten to reflect the manuscript. 

C2. Introduction reads like a Discussion. The authors describe vascular damage, then refer to reports of ED in OSAS, and refer to OSAS symptoms associated with ED. Introduction should be used to justify the present study, not just enlist prior studies in a random fashion.

C3. Some of the text is not in English. English language edits are required.

C4. Post CPAP AHI is 20.7 (8.9). Why is this so high? If CPAP treatment was not effectively administered, it will likely show inadequate results. 

C5. The included studies need to be better described. Suggest using the following for formatting- 

Baba, R. Y., Mohan, A., Metta, V. V. S., & Mador, M. J. (2015). Temperature controlled radiofrequency ablation at different sites for treatment of obstructive sleep apnea syndrome: a systematic review and meta-analysis. Sleep and Breathing, 19(3), 891-910.

C6. Discussion should highlight key findings, described potential mechanisms of ED in OSA, how therapy helps etc. It is very brief in its present format. 

C7. Results are poorly described.

C8. Key details such as protocol code are missing.